# *Adenophora triphylla* var. *japonica* Inhibits *Candida* Biofilm Formation, Increases Susceptibility to Antifungal Agents and Reduces Infection

**DOI:** 10.3390/ijms222212523

**Published:** 2021-11-21

**Authors:** Daseul Kim, Ki-Young Kim

**Affiliations:** 1Graduate School of Biotechnology, Kyung Hee University, Seocheon, Giheung, Yongin 446-701, Gyeonggi-do, Korea; charybde@naver.com; 2Department of Genetic Engineering, College of Life Science, Graduate School of Biotechnology, Kyung Hee University, Seocheon, Giheung, Yongin 446-701, Gyeonggi-do, Korea

**Keywords:** *Adenophora triphylla* var. *japonica*, biofilm formation, *Candida* spp., susceptibility

## Abstract

(1) Background: *Candida* is the most common cause of fungal infections worldwide, but due to the limited option of antifungal therapies, alternative strategies are required. (2) Methods: *Adenophora triphylla* var. *japonica* extract was used for the biofilm formation assay using RPMI1640. The combinatorial antifungal assay, the dimorphic transition assay, and the adherence assay were done to see the influence of inhibition of biofilm formation. qRT-PCR analysis were performed to check the gene expression. (3) Results: *Adenophora triphylla* var. *japonica* extract inhibited the *Candida* biofilm formation. Treatment of extract increased the antifungal susceptibility of miconazole from a 37% reduction in fungal growth to 99.05%, and also dose-dependently reduced the dimorphic transition of *Candida* and the attachment of *Candida* to HaCaT cells. The extract blocked the expression of hyphal-related genes, extracellular matrix genes, Ras1-cAMP-PKA pathway genes, Cph2-Tec1 pathway gene, and MAP kinase pathway gene. (4) Conclusions: In this study, the treatment of *Adenophora triphylla* var*. japonica* extract showed inhibition of fungal biofilm formation, activation of antifungal susceptibility, and reduction of infection. These results suggest that fungal biofilm formation is a good target for the development of antifungal adjuvants, and *Adenophora triphylla* var*. japonica* extract should be a good candidate for biofilm-associated fungal infections.

## 1. Introduction

*Candida albicans* is an opportunistic fungal pathogen that can cause systemic infections in individuals when mucosal barriers are disrupted, or the immune system is compromised [1]. Nosocomial *C. albicans* infection is often linked with the ability to form biofilms on mucosal surfaces and implanted medical devices [1,2,3]. 

Biofilm formation is a finely regulated process by multiple interconnected signaling pathways [4,5,6,7,8] and causes structured microbial communities to attach to surfaces [9,10]. The polysaccharide matrix indeed can act as a barrier for adherent cells by preventing the invasion of antifungal agents, conferring drug resistance. Adherent *C. albicans* cells living within the biofilm are up to 1000 times more resistant to common antifungal agents than planktonic cells, even in the absence of specific drug-resistance genes [11]. Thus, biofilms are a source of viable fungal cells that can potentially cause systemic infection with a mortality rate of 40–60% [12].

There have been many efforts to find novel natural compounds from medicinal plants to block biofilm-associated infections [13]. Medicinal plants are used in a variety of traditional remedies to treat cancer, infections, fever, asthma, and many other ailments, and generally have fewer side effects than over-the-counter medicines. Therefore, medicinal plants should be a new source of alternative therapies for the treatment of *Candida* infections [14,15].

*Adenophora triphylla* var. *japonica* (Korean name Zandae) has been used as an herbal medicine in Korea, China, and Japan to control lung diseases such as cough, sputum, asthma, and airway inflammatory diseases. Recently it was reported that extracts of *A. triphylla* var. *japonica* exhibited anti-inflammation, antidiabetic, antiparasitic, antitussive, and hepatoprotective effects [16,17,18]. 

In this study, *A. triphylla* var. *japonica* extract was tested for anti-biofilm-forming activity against several fungi including *C. albicans, C. tropicalis, C. glabrata,* and *C. parapsilosis*. Interestingly, the extract blocked biofilm formation, increased susceptibility to antibiotics, and reduced fungal infections.

## 2. Results

### 2.1. A. triphylla var. japonica Did Not Inhibit the Fungal Growth

Some antifungal agents inhibit fungal biofilm formation because they kill fungi and can indirectly reduce biofilm formation. *A. triphylla* var. *japonica* extract did not affect the growth of *C. albicans* after 24 h incubation. This result suggested that the extract’s inhibition of biofilm formation was not due to the reduced growth of *Candida* (Figure 1, Appendix A)

### 2.2. A. triphylla var. japonica Extract Inhibited the Biofilm Formation of Candida

Biofilms are particularly important for fungi to survive and infect. *A. triphylla* var. *japonica* extract was used to test whether it blocks fungal biofilm formation. *A. triphylla* var. *japonica* extract dose-dependently inhibited *Candida* biofilm formation in all strains tested (Table 1) with an IC_50_ value of approximately 6.25 µg/mL against *C. albicans* (Figure 1). The IC_50_ value of the other strains was 6.25 μg/mL (*C. parapsilosis* var. *parapsilosis and C. glabrata*), 25 μg/mL (*C. tropicalis* var. *tropicalis* and *C. tropicalis*), and 50 μg/mL (*C. parapsilosis*). (Figure 2, Table 1).

### 2.3. A. triphylla var. japonica Extract Increased the Susceptibility of Antifungal Agents against C. albicans

Biofilms are associated with antifungal activity; therefore, inhibition of biofilm formation is expected to increase antifungal efficacy. Treatment with 3.125 μg/mL of extract increased the susceptibility of miconazole from 38% inhibition to 99% inhibition of fungal growth. The extract also showed synergistic antifungal activity with magnoflorin (99% growth inhibition with 3.125 μg/mL extract compared with 13% by magnoflorin alone) and diosin (99% growth inhibition with 3.125 μg/mL extract compared with 23% by dioscin alone) (Figure 3a–c).

### 2.4. A. triphylla var. japonica Blocked the Dimorphic Transition from Yeast to Hyphae Form

The yeast-to-hyphae conversion is an important virulence factor for *C. albicans*. The formation of hyphae aids the subsequent invasive growth of *C. albicans*, infiltrating host tissues and causing systemic infection [6]. Extracts were tested to determine whether the dimorphic transition was affected and found that 1.56 μg/mL of extracts significantly inhibited dimorphic transition of 45% in RPMI1640 and 39% in 10% FBS YPD medium compare with untreated control (Figure 4).

### 2.5. A. triphylla var. japonica Extract Eeduced Fungal Adherence to the HaCaT Cells

Fungal biofilm formation for device-related infections is an important medical problem. *A. triphylla* var. *japonica* extract dose-dependently decreased the adhesion of *Candida* to human HaCaT cells by extract treatment (Figure 5).

### 2.6. A. triphylla var. japonica Extract did Not Affect the Growth of Human Originated Cell

The cytotoxic effect of *A. triphylla* var. *japonica* extract on HaCaT cells and macrophages THP-1 was tested by MTT assay. *A. triphylla* var*. japonica* extract did not show any significant cytotoxic effect on both HaCaT and THP-1 cells (Figure 6a,b).

### 2.7. A. triphylla var. japonica Extract Inhibited the Expression of Biofilm Formation and Infection-Related Genes

To understand the molecular basis of the inhibition of biofilm formation and reduction of *Candida* infection, *A. triphylla* var. *japonica* extracts were tested for the expression of genes involved in biofilm formation, hyphae growth, and cell adhesion by qRT-PCR. The expressions of biofilm formation-related genes: [*CAN2* (IC_50_ = 1.56 μg/mL), *EHT1* (IC_50_ = 1.56 μg/mL), *TPO4* (IC_50_ = 1.56 μg/mL), and *OPT7* (IC_50_ = 1.56 μg/mL)]; Ras1-cAMP-PKA pathway related genes: [*RAS1* (IC_50_ = 1.56 μg/mL), *EFG1* (IC_50_ = 1.56 μg/mL), *TEC1* (IC_50_ = 1.56 μg/mL), *HST7* (IC_50_ = 1.56 μg/mL), and *CYR1* (IC_50_ = 1.56 μg/mL)]; hyphal-specific genes: [*ALS3* (IC_50_ = 12.5 μg/mL), *ECE1* (IC_50_ = 12.5 μg/mL), and *HWP1* (IC_50_ = 3.125 μg/mL)]; and extracellular matrix-related genes: [*GSC1* (IC_50_ = 3.125 μg/mL), *ADH5* (IC_50_ = 1.56 μg/mL), *ZAP1* (IC_50_ = 3.125 μg/mL), and *CSH1* (IC_50_ = 3.125 μg/mL)] were dose-dependently reduced by treatment of *A. triphylla* var. *japonica* extract (Figure 7a–d).

## 3. Discussion

Many fungi, including *Candida*, live inside the human body, but when *Candida* starts to grow out of control, it can cause an infection known as candidiasis. In fact, *Candida* is the most common cause of fungal infections in humans [1,2,3]. Antifungals can cause side effects and resistance, and there are several reports of resistance, including *Candida auris*, an emerging problematic organism [1,2,3]. Outbreak response is complicated by limited treatment options and inadequate disinfection strategies. Therefore, new approaches with novel targets for antifungal agents are needed.

In this study, *A. triphylla* var. *japonica* extract exhibited biofilm formation inhibitory activity against all *Candida* species tested (Figure 2, Table 1). The mechanism of *Candida* biofilm formation slightly differs depending on the species [5,6,7,8,9,10,11,12,13,14,15,16,17,18,19,20,21,22,23,24,25,26,27,28,29]. Comparisons are difficult because all *Candida* species have similar regulatory genes at each stage of biofilm formation. However, it can be emphasized that the *ALS* gene is involved in the adhesion process in three *Candida* species (*C. albicans, C. parapsilosis, and C. tropicalis*), and that this phenomenon is regulated by *Epas* in *C. glabrata*. In addition, some transcription factors described as being involved in *C. albicans* biofilm formation (*BCR1, EFG1* and *HWP1*) are the same as those involved in *C. parapsilosis* biofilms. *C. glabrata* is a species that exhibits more contrast with respect to other *Candida* species, reflecting genetic distance [5]. *A. triphylla* var*. japonica* extract inhibits *Candida* biofilm formation, suggesting that there should be a common mechanism that is regulated by plant extract to influence the formation of biofilms among *Candida* species, but more studies are needed. In addition, *A. triphylla* var. *japonica* extract increased the antifungal activity of miconazole, magnoflorine, and dioscin (Figure 3), and decreased *Candida* infection (Figure 4 and Figure 5). These results showed that biofilm formation was associated with the susceptibility of antifungal agents and fungal infections.

According to gene expression analysis, the expression of genes related to biofilm formation, hyphae growth, and cell adhesion was significantly reduced by *A. triphylla* var. *japonica* extract treatment. Biofilm formation in *Candida* species is determined by various transcription factors, including *BCR1, EFG1, TEC1,* and *NDT80*, which act as components in multiple pathways, and their participation in *Candida* adhesion also suggests that although biofilm formation is initially tested, adhesion or infectious factors are also controlled by treatment with *A. triphylla* var. *japonica* extract. Further studies should be performed including the precise targeting of *A. triphylla* var*. japonica* extracts, effector components, and in vivo testing (Figure 1).

In conclusion, *A. triphylla* var*. japonica* extracts inhibited *C. albicans* biofilm formation, increased antifungal activity of antifungal agents, and decreased fungal adhesion to host cells. Therefore, *A. triphylla* var*. japonica* extract should be a good candidate for biofilm-forming fungal infections.

## 4. Materials and Methods

### 4.1. Fungal Strains Used in This Study

The *Candida* strains used in the study are listed in Table 2. *C. albicans* KCTC 7965 and ATCC 10231 are the same strain. *C. tropicalis* KCTC 7212 was sourced from a patient with bronchomycosis*. C. parapsilosis var. parapsilosis* KACC 45480 and ATCC 22019 are the same strain. *C. glabrata* KCTC 7219 was sourced from faces. *C. parapsilosis* KACC 49573 was sourced from Canis familialis. All strains were stored in 20% glycerol at −70 °C and incubated on YPD plates [Peptone 20 g/L (BD Difco, Franklin Lakes, NJ, USA), yeast extract 10 g/L (BD Difco, Franklin Lakes, NJ, USA) and 2% glucose (*w/v*) (Daejung, Korea)] [27,28,29,30].

### 4.2. Extraction of A. triphylla var. japonica

The leaves of *A. triphylla* var. *japonica* were obtained from Jeju Island. Three-hundred grams of the sample was extracted using 3 L distilled water at 80 °C for 8 h, concentrated at 40 °C using a rotary evaporator, and then freeze-dried. For the experiment, 10 mg of plant extract powder was dissolved in 1 mL of dimethyl sulfoxide [14,15,16,17,18,19].

### 4.3. Growth Curve Assay

Fungal cultures were prepared using fresh YPD at 1 × 10^6^ cell/mL of *C. albicans* [23,24,25,26,27,28,29]. One hundred µg/mL of *A. triphylla* var. *japonica* extract was added and incubated at 37 °C. Growth was assessed by measuring OD_600_ using a microplate reader after indicated time [23,24,25]. Experiments were performed in triplicate.

### 4.4. Biofilm Formation Assay

*A. triphylla* var. *japonica* extracts were prepared in 96-well plate plates (SPL, Korea) with the concentration ranging from 6.25 to 100 μg/mL. Wells without test compound served as controls (DMSO concentration 0.1%). *C. albicans* suspension at 1 × 10^6^ CFU/mL was prepared in RPMI 1640 medium [24,25,26,27,28,29,30,31]. Then, 100 μL of the solution was inoculated into a 96-well flat-bottom plate and incubated at 37 °C for 24 h with shaking. Biofilm formed was washed with PBS to remove nonadherent cells, and then 100 μL of 1% aqueous crystal violet was applied for 30 min. Each well was washed three times with PBS and destained with 150 μL of 30% acetic acid for 15 min. The absorbance was measured at 595 nm with a microplate reader (Bio Tek Instruments, Korea). Experiments were performed in triplicate.

### 4.5. Antifungal Activity Assay

*C. albicans* were grown overnight in YPD diluted to 1 × 10^6^ cells/mL. Induction of biofilm formation was performed as described above. The broth microdilution method was applied to determine the MICs using microplates [32]. Serial twofold dilutions of *A. triphylla var. japonica* extracts were prepared in YPD broth. The MIC value was determined as the minimum compound concentration. Miconazole (3.125 µg/mL), magnoflorin (3.125 µg/mL), and dioscin (3.125 µg/mL) alone or with the indicated concentration of *A. triphylla var*. japonica extract were added to each well and incubated for 24 h at 37 °C [23,24,25,26,27]. Experiments were performed in triplicate.

### 4.6. Dimorphic Transition Assay

*C. albicans* were grown overnight in YPD medium. 1 × 10^6^ cells/mL of *Candida* with or without extract were incubated in YPD medium supplemented 10% fetal bovine serum medium or sodium bicarbonate-free RPMI medium with glutamine buffered with MOPS [3-(N-morpholino) propanesulfonic acid] to pH 7 recommended by the CLSI standard to induce dimorphic transition for 4 h at 37 °C. Quantification of inhibition of dimorphic transition was performed by counting the number of yeast and hyphal cells in the population as previously described [25,26,27,28,29,30,31]. At least 1000 cells were counted in duplicate for each well and all assays were repeated five times. An image of the cells was obtained using a microscope.

### 4.7. Candida Adherence Assay

Human epithelial keratinocytes HaCaT cells were maintained in DMEM supplemented with 10% fetal bovine serum at 37 °C in a humidified atmosphere with 5% CO_2_. HaCaT cells (0.5 × 10^6^ cells per well) were grown to confluence for 24 h in 24-well plates.

1 × 10^6^ cells/mL of *C. albicans* mixed with indicated concentration of *A. triphylla* var. *japonica* extract was incubated at 37 °C for 24 h. After the DMEM was drained, the plate was carefully washed three times with PBS to remove non-adherent cells. The number of adherent *C. albicans* cells was measured by scraping the cells with a sterile scraper. Serial dilutions were plated on YPD agar to determine viability. A microscope was used to obtain representative images of the results [26,27].

### 4.8. MTT Assay

Cytotoxicity of *A. triphylla* var. *japonica* extracts against HaCaT and THP-1 was tested by a slightly modified MTT assay. 1 × 10^4^ HaCaT and THP-1 in DMEM and RPMI1640 media, respectively, were added to each well containing the indicated concentration of extracts and incubated for 24 h. After discarding the media, MTT (3-(4,5-dimethyl-thiazol-2-yl)-2,5-diphenyltetrazolium bromide, Sigma, St. Louis, MO, USA) in PBS was added into each well, followed by incubation for 3 h at 37 °C. The medium was then removed, and cells were suspended in 100 μL DMSO for 10m. Cell viability was calculated as optical density (OD_540_) values measured using a microplate reader (BioTek Instruments, Korea) and reported as a percentage compare with vehicle control [25]. Experiments were performed in triplicate.

### 4.9. qRT-PCR Analysis

*C. albicans* were grown overnight in YPD and diluted to 1 × 10^6^ cells/mL. The indicated concentration of *A. triphylla* var. *japonica* extract was incubated in RPMI1640 with shaking at 37 °C for 24 h. Total RNA was isolated using TRIzol reagent (Life Technology, Thermo Fisher Scientific, Waltham, MA, USA) according to the manufacturer’s instructions, and cDNA was obtained by reverse transcriptase (NanoHelix, Korea) reaction using 1 μg of RNA. qRT-PCR was performed with 2 × SybrGreen qPCR Mater Mix (CellSafe, Korea). Transcript levels of genes were calculated using the formula 2-ΔΔCT. The primer sequences used are listed in Table 3. ACT1 served as an internal control [23,24,25,26,27,28,29,30,31,32].

### 4.10. Statistical Analysis

All experiments were performed at least three times and data were presented as the ± mean S.D. Graphs were generated using Microsoft Excel (Microsoft Corp., Redmond, WA, USA).

## Data Availability

The data presented in this study are available on request from the corresponding author.

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
