# Peer review of "Adenophora triphylla var. japonica Inhibits Candida Biofilm Formation, Increases Susceptibility to Antifungal Agents and Reduces Infection"

_ijms, 2021, doi:10.3390/ijms222212523_

Round 1

Reviewer 1 Report

1 / Scheme No. 1 should summarize the results, not open them - the location of the scheme should be changed

2 / for Figure 1, a better form of result presentation should be found - a more compact one

3 / we do not know how the antifungal activity was measured (what type of test was used). If the method used was the same as for biofilm testing, no antifungal activity was tested but rather the activity of biofilm formation.

4 / the adherence assay is not completely described

5 / the order of the presented results is not proper. First, we should see how the extract affects the cell growth, then the adhesive properties, and finally the formation of biofilm

6 / no statistical significance  is included in the figures

7 / I think that the plant extract modifies the growth of yeast cells. Maybe it is worth testing it for all strains, to be able to present some general conclusion

8 / the conclusion included in the discussion section is incomprehensible: "Although the mechanism of Candida biofilm formation slightly differs depending on the species (no mechanism was indicated here), A. triphylla var. Japonica extract inhibits Candida biofilm formation suggesting that there should be a common mechanism to induce biofilms among Candida species” (contradiction).

9 / the data concerning the influence on gene expression are poorly used to explain the possible mechanism of the extract action.

The text requires a thorough linguistic correction.

Author Response

Response to Reviewer 1 Comments

Adenophora triphylla var. japonica inhibits Candida biofilm formation, increases susceptibility to antifungal agents and reduces infection.

We thank the editors and the reviewers for their thoughtful and helpful comments. We have addressed, in a point-by-point manner, all the suggestions and queries from the journal and the reviewer and marked with red in manuscript. The input from the reviewers has allowed us to improve the clarity and quality of our paper. We have included below our point-by-point response to the reviewers’ comments and have included these additions and alterations to the revised manuscript.

Comments and Suggestions for Authors

Point 1:  Scheme No. 1 should summarize the results, not open them - the location of the scheme should be changed

Response 1: Thanks for your suggestion. We changed the location for the part you mentioned and corrected it.

Point 2:  for Figure 1, a better form of result presentation should be found - a more compact one

Response 2: Thank you for your constructive suggestion. We modified the picture as you mentioned.

Figure 2. A. triphylla var. japonica extracts inhibited biofilm formation of C. albicans pp. Biofilm formation of C. albicans was induced in RPMI1640 with indicated concentrations of A. triphylla var. japonica extract for 24 hours at 37 °C. Experiments were performed in triplicate.

Table 1. A. triphylla var. japonica extracts inhibited biofilm formation of Candida spp IC50 value.

Strain

IC50

C. clbicans

6.25 µg/mL

C. tropicalis var. tropicalis

25 μg/mL

C. parapsilosis var. parapsilosis

6.25 μg/mL

C. glabrata

6.25 μg/mL

C. tropicalis

25 μg/mL

C. parapsilosis

50 μg/mL

Point 3: we do not know how the antifungal activity was measured (what type of test was used). If the method used was the same as for biofilm testing, no antifungal activity was tested but rather the activity of biofilm formation.

Response 3: Thank you for your constructive suggestion. We modified the word as you mentioned.

4.5. Combinatorial antifungal activity assay

  1. albicans grown overnight in YPD diluted to 1×106 cells/mL. Induction of biofilm formation was performed as described above. The broth microdilution method was applied to determine the MICs using microplates [32]. Serial twofold dilutions of A. triphylla var. japonica extracts were prepared in YPD broth. The MIC value was determined as the minimum compound concentration. Miconazole (3.125 µg/mL), magnoflorin (3.125 µg/mL) and dioscin (3.125 µg/mL) alone or with the indicated concentration of A. triphylla var. japonica extract was added to each well and incubated for 24 hours at 37 °C [23–27]. Experiments were performed in triplicate.

Point 4:  the adherence assay is not completely described

Response 4: Thank you for your professional advice. As you mentioned, we added some more experimental methods.

4.7. Candida adherence assay

Human epithelial keratinocytes HaCaT cells were maintained in DMEM supplemented with 10% fetal bovine serum at 37 °C in a humidified atmosphere with 5% CO2. HaCaT cells (0.5×106 cells per well) were grown to confluence for 24 hours in 24-well plates. 1×106 cells/mL of C. albicans mixed with indicated concentration of A. triphylla var. japonica extract was incubated at 37 °C for 24 hours. After the DMEM was drained, the plate was carefully washed 3 times with PBS to remove non-adherent cells. The number of adherent C. albicans cells was measured by scraping the cells with a sterile scraper. Serial dilutions were plated on YPD agar to determine the adherent bacterial cells. A microscope was used to obtain representative images of the results [26,27].

Point 5:  the order of the presented results is not proper. First, we should see how the extract affects the cell growth, then the adhesive properties, and finally the formation of biofilm

Response 5: Thank you for your careful work. As you mentioned, I changed the order of the test results.

Point 6:  no statistical significance is included in the figures

Response 6: Thank you for your valuable suggestions. We have changed the figures including SD

Point 7:  I think that the plant extract modifies the growth of yeast cells. Maybe it is worth testing it for all strains, to be able to present some general conclusion

Response 7: Thank you for your constructive suggestion. We do not have data on growth curves for all strains.

(A)

(B)

(C)

(D)

(E)

Figure S1. A. triphylla var. japonica extract did not inhibit the growth of Candida spp. (A) C. tropicalis var tropicalis, (B) C. parapsilosis var. parapsilosis, (C) C. glabrata, (D) C. tropicalis, and (E) C. parapsilosis containing 100 μg/mL of A. triphylla var. japonica extract was incubated for 24 hours at 30 °C. Experiments were performed in triplicate.

Point 8:  the conclusion included in the discussion section is incomprehensible: "Although the mechanism of Candida biofilm formation slightly differs depending on the species (no mechanism was indicated here), A. triphylla var. Japonica extract inhibits Candida biofilm formation suggesting that there should be a common mechanism to induce biofilms among Candida species” (contradiction).

Response 8: Thank you for your careful work. We have corrected the part you mentioned.

In this study, A. triphylla var. japonica extract exhibited biofilm formation inhibitory activity against all Candida species tested (Figure 2, Table 1). Although the mechanism of Candida biofilm formation slightly differs depending on the species [5, 27]. Comparisons are difficult because all Candida species have similar regulatory genes at each stage of biofilm formation. However, it can be emphasized that the ALS gene is involved in the adhesion process in three Candida species (C. albicans, C. parapsilosis, and C. tropicalis), and that this phenomenon is regulated by Epas in C. glabrata. In addition, some transcription factors described as being involved in C. albicans biofilm formation (BCR1, EFG1 and HWP1) are the same as those involved in C. parapsilosis biofilms. C. glabrata is a species that exhibits more contrast with respect to other Candida species, reflecting genetic distance [5]. A. triphylla var. japonica extract inhibits Candida biofilm formation suggesting that there should be a common mechanism that regulated by plant extract to influence the formation of biofilms among Candida species, but more studies are needed.  In addition, A. triphylla var. japonica extract increased the antifungal activity of miconazole, magnoflorine and dioscin (Figure 3), and decreased Candida infection (Figure 4 and 5). These results showed that biofilm formation was associated with the susceptibility of antifungal agents and fungal infections.

Point 9:  the data concerning the influence on gene expression are poorly used to explain the possible mechanism of the extract action.

Response 9: Thank you for your careful work. As you mentioned, I looked up the Candida biofilm formation mechanism.

Some experiments showed the biofilm formation inhibition but most of them showed gene expression change for the biofilm formation, I think that is not a perfect method to show the biofilm formation inhibition, because gene expression change is not perfectly match with the inhibition of signal transduction pathway. We next plan to do use the single or some compounds belong to the plant extract, and that time we will use some more candida species including C.auris and use some known signal transduction pathway inhibitors to check the inhibition by single compound, and after that try to do find the protein molecules which directly and initially influenced. So for now, we though the gene expression results showed general influence by plant extract even though that is not perfect.

Lingmei, S.; Kai, L.; Dayoung, W. Effects of magnolol and honokiol on adhesion, yeast-hyphal transition, and formation of biofilm by Candida albicans. PLos One 2015, Volume 10, e0117695.

Paola, M.; Roberta, F.; Cosmeri, R.; Tavanti, A.; Lupetti, A. Inhibition of Candida albicans biofilm formation by the synthetic Lactoferricin derived peptide hLF1-11. PLoS One 2016, Volume 11, e0167470.

Reviewer 2 Report

General thoughts:

The results presented in the paper have potential but need to be supplemented. Especially a discussion where virtually no literature is cited. The purpose of the discussion is to compare the results with the data published so far, but this is not the case here at all. In addition, the last paragraph of the introduction should be edited as it suggests that all analyzes were performed for all strains, while a significant portion of the results only concerns C. albicans.

Questions:

  1. Why was it chosen to choose miconazole, magnoflorine and dioscin? The text should explain the reasons for such a choice, indicating how they work and for which microorganisms they proved to be effective.
  2. Fig. 2. In the three panels, the third column has the same signature. Moreover, in addition to viability, have the type of morphological form and OD been checked? Perhaps the decreased survival rate is a more complex effect.
  3. What is the difference between the strains used, especially C. parapsilosis var. parapsilosis and C. parapsilosis? Such an explanation should be found in the paper. In Table 1, it would be good to include information from where the tested strains were isolated.
  4. The results presented in the paper concern the stage of biofilm formation. Were analogous results performed for the mature C. albicans biofilm? This should be analyzed also.
  5. MTT measurements were performed in DMEM and RPMI 1640 medium which contain phenol? Did it not affect the measurements?

Author Response

Response to Reviewer 2 Comments

Adenophora triphylla var. japonica inhibits Candida biofilm formation, increases susceptibility to antifungal agents and reduces infection.

We thank the editors and the reviewers for their thoughtful and helpful comments. We have addressed, in a point-by-point manner, all the suggestions and queries from the journal and the reviewer and marked with red in manuscript. The input from the reviewers has allowed us to improve the clarity and quality of our paper. We have included below our point-by-point response to the reviewers’ comments and have included these additions and alterations to the revised manuscript.

Point 1:  The results presented in the paper have potential but need to be supplemented. Especially a discussion where virtually no literature is cited. The purpose of the discussion is to compare the results with the data published so far, but this is not the case here at all. In addition, the last paragraph of the introduction should be edited as it suggests that all analyzes were performed for all strains, while a significant portion of the results only concerns C. albicans.

Response 1: Thank you so much. We have added more references as you mentioned.

Araújo, D.; Henriques, M.; Silva, S. Portrait of Candida species biofilm regulatory network genes. Trends Microbiol 2017, Volume 25, 62–75.

Cavalheiro, M.; Teixeira, M.C. Candida biofilms: threats, challenges, and promising strategies. Front Med (Lausanne) 2018, Volume 5, 28.

Questions:

Point 2:  Why was it chosen to choose miconazole, magnoflorine and dioscin? The text should explain the reasons for such a choice, indicating how they work and for which microorganisms they proved to be effective.

Response 2:  Thank you for your professional advice. The reason miconazole, magnoflorine, and dioscin were selected as antifungal agents was to determine whether these three substances are known as antifungal agents, and in the case of miconazole, the antifungal mechanism is well known and the other 2 substances have potential to be used as a back bone for new antifungal agent. So we want to know the mechanism of antifungal agents is important to increase the antifungal activity by inhibition of biofilm formation.

Point 3:  Fig. 2. In the three panels, the third column has the same signature. Moreover, in addition to viability, have the type of morphological form and OD been checked? Perhaps the decreased survival rate is a more complex effect.

Response 3: Thank you very much. It has been corrected in the picture section. We did not measure the viability of cells by OD but measured the number of cells by inoculating them on an Agar plate.

: (a)

(b)

(c)

Point 4:  What is the difference between the strains used, especially C. parapsilosis var. parapsilosis and C. parapsilosis? Such an explanation should be found in the paper. In Table 1, it would be good to include information from where the tested strains were isolated.

Response 4:  Thank you for your professional advice. We have added a separate reference for the part you mentioned.

4.1. Fungal strains used in this study

The Candida strains used in the study are listed in Table 2. C. albicans KCTC 7965 and ATCC 10231 was the same strain. C. tropicalis KCTC 7212 was sourced from a patient with bronchomycosis. C. parapsilosis var. parapsilosis KACC 45480 and ATCC 22019 are the same strain. C. glabrata KCTC 7219 was sourced from faces. C. parapsilosis KACC 49573 was sourced from Canis familialis.  All strains were stored in 20% glycerol at -70 °C and incubated on YPD plates [Peptone 20 g/L (BD Difco, Belgium), yeast extract 10 g/L (BD Difco, Belgium) and 2% glucose (w/v) (Daejung, Korea)] [30].

Point 5:  The results presented in the paper concern the stage of biofilm formation. Were analogous results performed for the mature C. albicans biofilm? This should be analyzed also.

Response 5: Thank you very much. We though that the plant extract inhibit the biofilm formation, but could not influence the mature biofilm, the results that we have showed the amounts of biofilm was not changed by treatment of plant extract.

Point 6:  MTT measurements were performed in DMEM and RPMI 1640 medium which contain phenol? Did it not affect the measurements?

Response 6: Thank you very much. We have added more explanations about the experimental method.

4.8. MTT assay

Cytotoxicity of A. triphylla var. japonica extracts against HaCaT and THP-1 was tested by a slightly modified MTT assay. 1×104 HaCaT and THP-1 in DMEM and RPMI1640 me-dia, respectively, were added to each well containing indicated concentration of extracts and incubated for 24 hours. After discard the media, MTT (3-(4,5-dimethyl-thiazol-2-yl)-2,5-diphenyltetrazolium bromide, Sigma, St. Louis, MO, USA) in PBS was added into each well, followed by incu-bation for 3 h at 37 °C. The medium was then removed, and cells were suspended in 100 μL DMSO for 10m. Cell viability was calculated as optical density (OD540) values meas-ured using a microplate reader (BioTek Instruments, Korea) and reported as a percentage compare with vehicle control [25]. Experiments were performed in triplicate.

Round 2

Reviewer 2 Report

In my opinion, the authors did not address all the comments. In response to question 2, they did not explain the mechanism of action of miconazole. Moreover, they did not justify the choice of drugs in the main text.

Why was it chosen to choose miconazole, magnoflorine and dioscin? The text should explain the reasons for such a choice, indicating how they work and for which microorganisms they proved to be effective.

In addition, gene names are written in capital letters and italics (line 195). The results presented in the supplements do not contain a caption.